# Does prior knowledge of food fraud affect consumer behavior? Evidence from an incentivized economic experiment

Syed Imran Ali Meerza[1]*, Christopher R. Gustafson[2]

**1** Department of Agriculture, Arkansas Tech University, Russellville, Arkansas, United States of America,
**2** Department of Agricultural Economics, University of Nebraska-Lincoln, Lincoln, Nebraska, United States of America

* smeerza@atu.edu

**Data Availability Statement:** All relevant data are in the manuscript and its Supporting Information file.

**Funding:** Financial support for the research came from Dr. Gustafson's Start-Up Research Fund,

## Abstract

This study uses a laboratory experiment to examine whether prior knowledge of food fraud persistently affects consumer behavior. We invited regular consumers of olive oil to participate in an olive oil valuation experiment. We used a within-subject design to compare consumers' willingness to pay (WTP) for Italian extra virgin olive oil (EVOO) before and after receiving information about labeling scandals in the Italian olive oil industry. After the first round of bidding, but before introducing information about labeling scandals or otherwise mentioning food fraud, we surveyed participants about whether they had heard of food fraud. Results indicate that prior knowledge of food fraud plays an important role in explaining consumers' valuation behavior, both in the pre-information baseline bidding and in how they update their valuation in response to information about a food fraud scandal. Consumers who reported prior knowledge of food fraud partially accounted for the possibility of food fraud in their initial pre-information valuation, submitting significantly lower bids than participants who did not report prior knowledge. They also reacted less to olive oil fraud information than consumers who reported no prior knowledge of food fraud. Findings of this study highlight the potential long-term consequences of increasing consumer awareness of food fraud incidents on consumer WTP for products in industries that have experienced food fraud scandals.

## Introduction

In recent years, food fraud scandals have gained widespread attention. Some, such as the adulteration of Chinese milk with melamine, have had serious health consequences. The adulteration of milk with melamine led to the deaths of at least six infants and the hospitalization of over 50,000 [1]. Others, including the discovery of horsemeat in many European meat products, and the mislabeling of Italian olive oils, result in consumers receiving an inferior product than what they paid for. These scandals highlight the vulnerability of the food system to intentional adulteration or misrepresentation of products based on economic motives [2] [3] [4].

which was provided by IANR, UNL. The funder had no role in study design, data collection and analysis, decision to publish, or preparation of the manuscript.

**Competing interests:** The authors have declared that no competing interests exist.

The economic costs of a food fraud scandal can range from lost sales and bankruptcies to adverse health consequences. For example, the total cost of the 2008 melamine milk scandal was estimated to be 10 billion dollars, which included the costs associated with product recalls and withdrawals, incident investigation, lost sales, decreases in shareholder value, and adverse health consequences [1].

One of the principal concerns about food fraud is that it will lead consumers to develop a baseline level of distrust in food product labeling [5], which will continue to affect consumer behavior long after a documented incident of food fraud occurs. Interest in this question has been heightened by increases in scientific and media attention to food fraud incidents. According to the United States Pharmacopeial Convention (USP), which monitors food fraud incidents, the total number of food fraud incidents in the two years from 2011 to 2012 was 60 percent as high as in the three decades between 1980 and 2010, while media coverage of food fraud incidents were nearly 80 percent as high [6] [7]. As more consumers become exposed to regular reports about food fraud incidents, it may affect their subjective perception that food products are mislabeled, leading them to maintain behaviors that they adopted to avoid suspected fraudulent products over long periods of time that previously only would have occurred in response to a specific food fraud incident.

Despite growing evidence of the widespread occurrence of food fraud, there is a paucity of empirical work documenting the impacts of fraudulent producer behavior in food markets on consumer decision-making. The empirical research that does exist is either experimental, with researchers directly exposing participants to information [8] [9], or tied to specific mislabeling events [10]. For instance, [10] use scanner data to examine German consumer behavior before and after a food fraud event that received extensive media attention in Germany. In the experimental literature, [8] studied hypothetical ready-to-eat meal choices of consumers across six European countries after the horsemeat scandal that occurred in Europe in 2012 in an online experiment and [9] estimated consumer response to exposure to food fraud information in an incentivized economic valuation experiment. A recently published study surveyed consumers on their opinions about food fraud [11]. In each of these studies, authors document that consumers are highly concerned about food fraud. [11] find that consumers develop strategies to avoid fraudulent food products and [9] results even suggest that food fraud information specific to products from one country spills over to products from other, unimplicated countries.

In this research, we address the effect of prior exposure to information about food fraud on consumer behavior in an economic valuation experiment. An experiment offers researchers the opportunity to generate data that would be difficult or impossible to obtain from secondary sources. To examine whether individuals begin to behave differently after exposure to food fraud incidents, it is vital to have a measure of individuals' prior exposure to food fraud, which would typically be an unobserved variable in data generated in real-world settings. For instance, in the study by [10], which used supermarket scanner data, the authors constructed an index of media coverage of the relevant food fraud incident because no information was available about whether each individual whose decisions were captured in the data had seen information about the food fraud event. The experimental setting, on the other hand, permits the researchers to directly elicit data on participants' prior exposure to information about food fraud incidents.

This valuation experiment features two rounds of elicitation of participants' willingness to pay (WTP) for a food product that has frequently been identified in food fraud articles: extra virgin olive oil (EVOO). When the first round of WTP elicitation occurred, participants had received no indication that food fraud would be a topic of interest in the study. After participants submitted their valuation in the first round, they completed a short survey that included questions about prior knowledge of any—not solely EVOO-related—food fraud incidents.

Upon completion of this survey, participants then read a short article about EVOO mislabeling incidents. Next, participants submitted their valuation for EVOO again.

If exposure to food fraud information influences consumers persistently, we expect to observe two patterns in consumer behavior, which would be observable in data generated in a valuation experiment involving rounds of bidding before and after the receipt of information about food fraud. These expected patterns constitute the two hypotheses we examine in this article. Our first hypothesis is that consumers with prior knowledge of food fraud value the EVOO less on average than consumers without prior knowledge of food fraud at baseline (that is, in the absence of specific food fraud information). Our second hypothesis is that participants with prior knowledge of food fraud respond less to the information about food fraud.

We examine differences in behavior between individuals with prior knowledge of food fraud, or *knowledgeable* consumers, and those without prior knowledge of food fraud, *unknowledgeable* consumers, in two ways to evaluate whether previous exposure to food fraud information affects long-term consumer valuation of food products. First, we compare the valuation of EVOO for knowledgeable and unknowledgeable participants before they had received any indication that the experiment involved mislabeling or food fraud. Because participants had not received any materials that even mentioned food fraud, differences in valuation between knowledgeable and unknowledgeable participants, controlling for variation in individual characteristics, should be driven by differences in the consumers' own background knowledge and subjective estimates of the probability that food is fraudulently labeled. Second, we examine how knowledgeable and unknowledgeable participants update their WTP after receiving information about food fraud in EVOO markets. We expect that knowledgeable participants should change their WTP less than unknowledgeable participants.

## Experimental design and procedure

We received clearance from the University of Nebraska-Lincoln (UNL) Institutional Review Board to conduct this research (20170616958 EX). The design of the experiment did not include deception of research subjects. The experiment was conducted in the Experimental and Behavioral Economics Lab at the University of Nebraska-Lincoln between September and November 2017. Since olive oil is one of the food categories most vulnerable to food fraud (Johnson 2014) and incidents of Italian olive oil fraud have been well documented [12] [13], Italian EVOO was chosen for this study. We recruited olive oil consumers by posting flyers in supermarkets and specialty food stores that provided information about how to register to participate in the experiment. A total of 107 olive oil consumers participated in this study. Written consent from study participant was obtained after explaining the laboratory experiment procedure. Each participant received a $30 participation fee, which was paid in cash when the experiment was completed. The experiment was programmed and delivered using Qualtrics (https://www.qualtrics.com/).

To test our hypotheses about persistent changes in consumers' expectations about the veracity of food product claims due to previous exposure to food fraud information leading to behavior change, we designed a multi-stage laboratory valuation experiment. After researchers led participants through an explanation of how the laboratory valuation experiment worked and a practice experiment with candy, participants valued Italian EVOO in a binding experimental auction. The first round of value elicitation occurred prior to any mention of food fraud or mislabeling.

After participants had submitted their valuations, they completed a short survey, which included a question about whether participants had been exposed previously to information about food fraud: "Had you heard about food fraud before coming to the study today?" If

exposure to food fraud information causes persistent changes in consumer valuation of products, knowledgeable consumers should submit lower pre-information bids for the EVOO. Next, all participants read a short article about incidents of mislabeling in the Italian olive oil market (see Appendix). At this point in the experiment, all participants had been exposed to information about food fraud, but the effect of the information provided in the experiment differs by consumer type. For unknowledgeable consumers, the text provides novel information about the existence of food fraud, while for knowledgeable consumers, the particular information—mislabeling scandals in the Italian EVOO industry—may be new, but the existence of food fraud is not. After reading the text about Italian EVOO mislabeling, participants then completed a second round of valuation of EVOO. Changes in valuation bids between the first and second round provide data to test our second hypothesis. Because the EVOO mislabeling text affects knowledgeable and unknowledgeable consumers' information sets differently, we expect that knowledgeable consumers' bids will change less than unknowledgeable consumers' bids in the second round of bidding.

We used the demand-revealing Becker-DeGroot-Marschak (BDM) mechanism [14] to elicit consumers' valuation of Italian EVOO. In the BDM mechanism, subjects submit a bid, representing their maximum WTP, for a good that is presented to them by the experimenter. After the subject has submitted their bid, an "experiment" price is randomly drawn from a distribution of prices. If the bid is higher than the randomly drawn experiment price, the participant purchases the good but pays only the experiment price. If the bid is lower than the experiment price, the participant does not purchase the good. Therefore, the amount the subject pays is independent of their bid, creating an incentive to value the product truthfully [14]. One of the advantages of the BDM mechanism is that it eliminates the possibility of bid affiliation among participants [15]. Bid affiliation describes a situation in which participants' bids become increasingly similar over multiple rounds of an experimental auction. While in other experimental auctions, such as the Vickrey Auction [16] or random nth price auction [17], participants' learn the winning bid—which identifies part of the bid distribution, the random experiment price in the BDM mechanism obviates bid affiliation.

Since the main objective of this study is to evaluate long-term effects food fraud events on consumer behavior by examining differences in the WTP of both consumers with prior knowledge and consumers without prior knowledge of food fraud across two informational conditions, avoiding bid affiliation is important. The threat of bid affiliation is particularly pronounced when some participants are unfamiliar with (or uninformed about) a product or attribute [18], though other researchers have documented bid affiliation even in experimental auctions for familiar products [15].

To avoid bid reduction in multi-unit auctions, at the end of the experiment, one round and one EVOO was randomly selected for each participant. The participant's WTP for that EVOO was compared to the randomly drawn experiment price to determine whether the participant would purchase the EVOO or not.

To account for potential differences in response to information about food fraud at different price levels, participants valued a high-priced and a low-priced Italian EVOO. The shelf prices of the high- and low-priced Italian EVOOs were $29 and $9, respectively, but participants were informed of a range of prices in which the bottles were sold. Participants were told that one EVOO was sold in the $25–30 price range, while the other was sold in the $5–10 price range. The size of each EVOO bottle was 500 ml (see Table 1).

Once all of the participants in a session had arrived at the laboratory, a researcher explained the experimental procedure. After completing a practice round to familiarize participants with the valuation mechanism and the computer interface, the experiment began. In the first round, participants were asked to submit their maximum WTP for the two different bottles of

**Table 1. Descriptive information of olive oil used in the experiment.**

| Type of EVOO | Bottle Size | Shelf Price |
|---|---|---|
| Italian High-priced EVOO | 500 ml | $29 |
| Italian Low-priced EVOO | 500 ml | $9 |

Notes: Prices of Italian EVOO reflect non-sale shelf prices observed in the study area during the data collection period.

Italian EVOO. Note that at this stage participants had not received any information about food fraud. Participants viewed the images of high- and low-priced Italian EVOO bottles on the computer screen. The country of origin, bottle size, and shelf price range were displayed alongside the images of the bottles. The EVOO bottles featured in the experiment were displayed in a random order to eliminate order effects. After bidding on both bottles of EVOO, participants completed a short survey on prior knowledge of food fraud and their perceptions of food fraud. This short survey was conducted after participants submitted their first-round bids in order to avoid unintentionally priming them to think more about the possibility that the products in the experiment might be mislabeled than they would under normal conditions.

Prior to the second round of bidding, participants read an article about the Italian olive oil industry and mislabeling scandals. After reading information about food fraud in the Italian olive oil industry, participants again submitted bids for the same set of Italian EVOO bottles. This was followed by surveys regarding demographic characteristics (e.g., age, gender, income, and education level) and olive purchasing behavior (e.g., types of olive oil purchased, quantity consumed per month, and average price spent per bottle). After the completion of the surveys, the random experiment price and EVOO bottle were drawn to determine the outcome of the experiment for each subject.

## Descriptive statistics and estimation strategy

Slightly more than one-third (36 percent) of participants reported having prior knowledge of food fraud, while 64 percent of participants reported no knowledge of food fraud. Table 2 summarizes data on participants' pre- and post-information WTP for Italian EVOO and their prior knowledge of food fraud. The unconditional (i.e., pooled across price levels) mean WTP for Italian EVOO was approximately $10. Participants' mean WTP for high-priced Italian EVOO was $18.54 before receiving information about Italian olive oil fraud; the average WTP decreased to around $10.36 after consumers were exposed to the information about Italian

**Table 2. Summary statistics of willingness to pay for extra virgin olive oil.**

| Variable | Mean | SD | Min | Max |
|---|---|---|---|---|
| WTP overall | 10.09 | 7.76 | 0 | 32 |
| High-Priced EVOO | | | | |
| *Before receiving olive oil fraud information* | 18.54 | 7.60 | 3 | 32 |
| *After receiving olive oil fraud information* | 10.36 | 7.01 | 0 | 26 |
| Low-Priced EVOO | | | | |
| *Before receiving olive oil fraud information* | 7.53 | 3.14 | 1.2 | 15 |
| *After receiving olive oil fraud information* | 3.91 | 2.89 | 0 | 12 |

Source: Data from the experiment

olive oil fraud. Mean WTP for low-priced Italian EVOO fell from $7.53 before information about food fraud to $3.91 afterwards.

Table 3 shows the summary statistics of participants' demographic characteristics, olive oil consumption habits, and perceptions of the frequency of food fraud occurrence. We examine these variables separately for consumers with and without prior knowledge of food fraud. Table 3 also reports the Fisher's Exact Test to investigate whether consumers with prior knowledge of food fraud (knowledgeable) are similar to consumers without prior knowledge (unknowledgeable) in terms of demographic characteristics and olive oil consumption habits. The composition across the groups *knowledgeable* and *unknowledgeable* is unbalanced only in terms of age (p = 0.02). There is also a marginally significant difference in olive oil consumption between knowledgeable and unknowledgeable consumers (p = 0.07). These two groups are balanced in terms of all other individual characteristics (Table 3).

To examine the role of prior knowledge of food fraud in consumers' baseline WTP for EVOO and response to food fraud information, this study employs regression techniques that make use of the panel structure of this dataset. We regress consumers' WTP for Italian EVOO on the dummy variables that capture differences in information, price level, and prior knowledge while taking into account control variables, including demographic characteristics, perceptions of the frequency of food fraud, and olive oil purchasing behavior:

$$WTP_{it} = \beta_0 + \beta_1\ Inf\_fraud_t + \beta_2\ Highpriced\_EVOO_{it} + \beta_3\ Highpriced\_EVOO_{it} * Inf\_fraud_t \\ + \beta_4\ Knowledgeable_i + \beta_5\ Knowledgeable_i * Inf\_fraud_t + \mathbf{X}_i'\boldsymbol{\theta} + \varepsilon_{it}\ (1)$$

where $WTP_{it}$ denotes the WTP of participant $i$ observed at time $t$ (t = 0 represents the pre-food fraud information stage; t = 1 represents the post-food fraud information stage). $Inf\_fraud_{it}$ is a dummy variable that captures the effect of receiving information about Italian olive oil fraud on WTP. The dummy variable $Highpriced\_EVOO_{it}$ identifies high-priced bottles of EVOO (1 = high price; 0 = low price), while $Knowledgeable_i$ captures participants with prior knowledge of food fraud (1 = prior knowledge; 0 = no prior knowledge). Further, $\mathbf{X}_i$ is a vector of participant characteristics and $\varepsilon_{it}$ is an i.i.d. standard normal error term.

The panel regression model in Eq 1 includes four observations for each of the 107 participants bidding for Italian EVOO bottles (both high- and low-priced) before and after receiving information about Italian olive oil fraud, resulting in 428 total bids. Since the laboratory experiment is designed in such a way that all participants are exposed to the same information treatment, and the values of the variables related to consumers' characteristics do not change across time (i.e., these variables are time-invariant), the random effects (RE) panel specification is appropriate. We use standard errors that are cluster-corrected at the individual level.

## Results

Table 4 presents the results of the panel regression model analyzing the effect of prior knowledge of food fraud on consumers' response to food fraud incidents. Regression results show that the WTP of knowledgeable consumers is $2.09 lower than the WTP of unknowledgeable consumers in the round of bidding that occurred before participants were exposed to food fraud information. This result is statistically significant at the 1% level. After receiving information about Italian olive oil fraud, unknowledgeable consumers' valuations for low-priced Italian EVOO decreased by $4.42, which is statistically significant at the 1% level, while their WTP for high-priced Italian EVOO fell by $9.07. While both knowledgeable and unknowledgeable participants decreased their valuations after being exposed to information, knowledgeable participants reacted less to the information. Unknowledgeable participants' bids fell

**Table 3. Summary statistics of control variables.**

| Variables | Group | | Total (N = 107) | p-value |
|---|---|---|---|---|
| | *Unknowledgeable (N = 69)* | *Knowledgeable (N = 38)* | | |
| Gender | | | | 0.54 |
| *Male* | 38% | 45% | 40% | |
| *Female* | 62% | 55% | 60% | |
| Education | | | | 0.26 |
| *Graduate Degree* | 25% | 39% | 30% | |
| *Bachelor's degree* | 19% | 24% | 21% | |
| *Associate Degree/Some College* | 27% | 16% | 23% | |
| *No College* | 29% | 21% | 26% | |
| Age | | | | 0.02** |
| *19–35 years* | 94% | 76% | 88% | |
| *36–49 years* | 4% | 19% | 9% | |
| *50+* | 2% | 5% | 3% | |
| Income | | | | 0.14 |
| *<60,000* | 68% | 63% | 67% | |
| *60,000–99,999* | 6% | 16% | 9% | |
| *100,000 and above* | 7% | 13% | 9% | |
| *Prefer not to answer* | 19% | 8% | 15% | |
| Olive Oil Consumption (monthly) | | | | 0.07* |
| *Greater than 1 liter* | 6% | 16% | 9% | |
| *≤1 Liter* | 94% | 84% | 91% | |
| Types of Olive Oil Purchased | | | | 0.39 |
| *EVOO* | 64% | 74% | 67% | |
| *Other Types but Not EVOO* | 36% | 26% | 33% | |
| Price Paid Per Bottle (on average) | | | | 0.49 |
| *≤ 10$* | 54% | 50% | 52% | |
| *10$–20$* | 42% | 50% | 45% | |
| *20$≥* | 4% | 0% | 3% | |
| Participant Perceptions % of mislabeling | | | | 0.16 |
| *0–50%* | 97% | 89% | 94% | |
| *51–75%* | 3% | 8% | 5% | |
| *75%≥* | 0% | 3% | 1% | |
| Participant Perceptions % of Domestically (U.S.) Produced Food Products Tested by US Food and Drug Administration (FDA) | | | | 0.90 |
| *0–50%* | 50% | 52% | 50% | |
| *51–75%* | 31% | 32% | 31% | |
| *75%≥* | 19% | 16% | 19% | |
| Participant Perceptions % of Imported Food Products Tested by US Food and Drug Administration (FDA) | | | | 0.78 |
| *0–50%* | 61% | 61% | 61% | |
| *51–75%* | 22% | 26% | 23% | |
| *75%≥* | 17% | 13% | 16% | |

Note: The reported *p*-values are from Fisher's Exact Test.

**: p≤5%,

*: p≤10%.

**Table 4. Effects of prior knowledge of fraud on consumers' response to food fraud information.**

| Independent variables | WTP for Italian EVOO |
|---|---|
| Inf_fraud (1,0) | -4.418*** (0.456) |
| Highpriced_Evoo (1,0) | 11.016*** (0.622) |
| Highpriced_Evoo* Inf_fraud | -4.565*** (0.550) |
| Knowledgeable | -2.089** (1.077) |
| Knowledgeable * Inf_fraud | 2.269*** (0.900) |
| **Olive Oil Consumption Behavior** | |
| Olive Oil Type (1,0) | -0.369 (0.891) |
| Liters Consumed per Month | -0.0004 (0.0001) |
| Price Paid per Bottle (on average) | 0.279** (0.122) |
| **Perceptions of Food Fraud** | |
| Percentage of Mislabeling | 0.032 (0.023) |
| Percentage of Domestically (U.S.) Produced Food Products Tested by FDA | 0.001 (0.016) |
| Percentage of Imported Food Products Tested by FDA | 0.029 (0.20) |
| **Demographic Characteristics** | |
| Gender (1,0) | -0.982 (0.802) |
| Age | 0.064 (0.086) |
| Education (reference: no college education) | |
| Graduate Degree | -1.594 (1.390) |
| Bachelor's degree | -0.916 (1.167) |
| Associate Degree/Some College | -0.486 (1.142) |
| Income (reference: prefer not to answer) | |
| <60,000 | 0.433 (1.033) |
| 60,000–99,999 | 1.486 (1.700) |
| 100,000 and above | 2.461 (1.644) |
| Constant | 2.584 (2.879) |
| $R^2$ | 0.53 |
| Wald $\chi^2$ | 574.90 |
| Number of observations | 428 |

Note:

***: $p \leq 1\%$,

**: $p \leq 5\%$,

*: $p \leq 10\%$.

by $2.27 more than knowledgeable participants' bids, which is statistically significant at the 1% level.

Fig 1 depicts mean estimated WTP averaged across low and high-priced EVOOS of participants with and without prior knowledge of food fraud before and after receiving Italian olive oil fraud information. As depicted in Fig 1, before receiving any information about Italian olive oil fraud, participants who reported prior knowledge of food fraud were willing to pay, on average, $1.60 less for Italian EVOO than participants without prior knowledge of food fraud. This result is statistically significant at the 5% level. However, after receiving information about Italian olive oil fraud, there is no statistically significant difference in WTP of participants with and without prior knowledge of food fraud. In other words, before being treated with Italian olive oil fraud information, knowledgeable consumers bid significantly less than unknowledgeable consumers. After being treated with Italian olive oil fraud information, there is no meaningful difference in WTP. Therefore, Fig 1 further reinforces our key results that

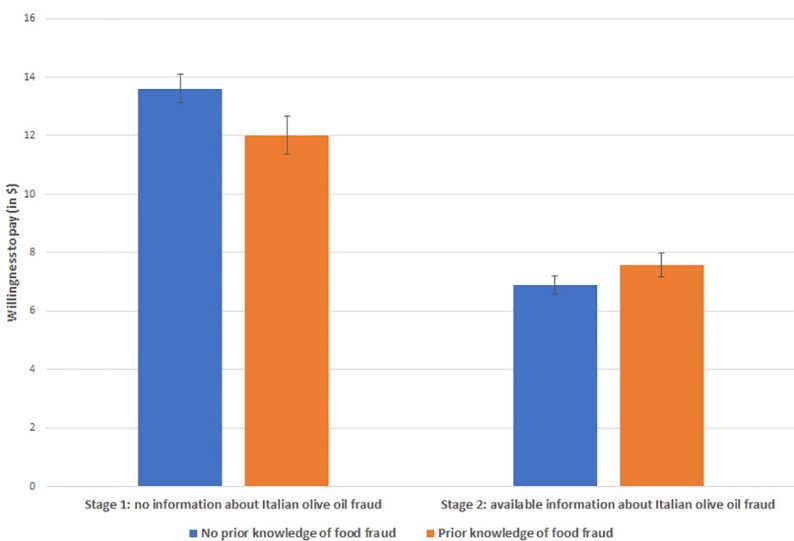

**Fig 1. Mean estimated WTP of participants with and without prior knowledge of food fraud before and after receiving information about mislabeling scandals in the Italian EVOO industry, averaged across low-price and high-price-range EVOOs with 95% confidence intervals.**

prior exposure to information about food fraud is an important factor in explaining consumer behavior in the presence of food fraud.

## Discussion and conclusions

This laboratory valuation experiment examines the role of prior knowledge of food fraud on knowledgeable (those with prior knowledge) and unknowledgeable (those without prior knowledge) consumers' baseline valuation of products—perhaps due to persistent, long-term changes in their subjective probability of products being mislabeled—and on consumers' response to food fraud information. We use participants' bids in the first round, before any mention was made of food fraud or mislabeling, to examine differences in baseline valuation of EVOO between knowledgeable and unknowledgeable consumers. We then examine changes in bids after participants read a text about food fraud incidents in the Italian olive oil industry to evaluate how knowledgeable and unknowledgeable consumers respond to information.

The analyses of the data show that before receiving information about Italian olive oil fraud, knowledgeable participants' WTP for Italian EVOO is significantly lower than unknowledgeable participants' WTP. This result indicates that knowledgeable participants (i.e., consumers with prior knowledge of food fraud) partially account for the possibility of food fraud in their initial bids. The analysis also shows that after receiving information about Italian olive oil fraud, participants reduced their valuation of Italian EVOO, irrespective of their prior knowledge of food fraud. However, unknowledgeable participants decreased their valuation more than knowledgeable participants after exposure to Italian olive oil fraud information, indicating that knowledgeable participants are less responsive to new information about food fraud incidents. Both of these patterns are consistent with food fraud information having persistent, long-lasting effects on individuals' perceptions of the credibility of producer claims about credence or experience attributes. Our findings suggest that knowledgeable participants have already changed their behaviors to account for the possibility that olive oil—a product that has had many widely publicized mislabeling scandals (e.g., [12], [13], [19])—is

fraudulently labeled and therefore react less to Italian olive oil fraud information than unknowledgeable participants.

Do consumers with prior knowledge of food fraud incidents behave differently? Since the intensity and frequency of food fraud have been on the rise and the media cover these scandals widely [7], the importance of this question has grown significantly. Our study finds that knowledgeable participants accounted for the possibility of food fraud before they had received any indication that the experiment involved mislabeling or food fraud. This behavior suggests that their baseline level of distrust in food product labeling differs from unknowledgeable participants. Moreover, they were less reactive to food fraud incidents since they already considered this possibility in their initial bids. Results of this study suggest that exposure to regular reports about food fraud may affect consumers' subjective perception of the accuracy of food labels. As a result, in the presence of food fraud scandals, consumers with prior knowledge of food fraud behave differently than consumers without prior knowledge of food fraud.

The pattern of results suggests that the increasing amount of media attention on food fraud incidents—potentially resulting in a growing percentage of the population internalizing the possibility of food fraud—could lead to the lemons problem [20]. If consumers exposed to information about food fraud incidents come to distrust product labeling, their valuation of these products is likely to decrease, which may mean that higher quality products will not be able to compete in the market if producers are unable to effectively signal that quality to consumers. While a solution to the asymmetric information problem already nominally exists in the market for EVOO—that is, quality certification, food producers are able to fraudulently label their products due to a lack of monitoring and oversight by the organizations responsible for certification systems, which may be governmental or private entities. Increasing the amount of funding available to monitor for mislabeled or adulterated products may be necessary to reduce the number of incidents of food fraud and maintain consumer trust in food labeling systems.

This study also contributes to a growing area of research that examines the effect of individuals' home-grown knowledge on economic decision-making. Most of this literature examines the effect of consumer knowledge in markets that feature informational complexity, such as branded vs. unbranded products (e.g. [21]), investment decisions ([22]; [23]), or wine ([24] [25]). For instance, [2]find that high-knowledge consumers respond more to objective information about wine than low-knowledge consumers. While our results are contrary to [25]—knowledgeable consumers in our study respond less to information, having apparently already accounted partially for the information—there is an important distinction. In the current research, consumers' prior knowledge about fraudulent food labeling appears to spill over to other food products—an effect documented in [9] in the context of decreases in consumer valuation of olive oils from different countries in response to negative information about one country—while [25] examined consumers' changing valuation for wine in response to receipt of objective information commonly provided on wine labels.

There are a few limitations to our study. As this research represents a first pass at examining whether previously encountered food fraud information influences consumer behavior, we do not have data on certain variables that would allow us to more accurately describe the relationship between exposure to food fraud information and subsequent consumer behavior. For instance, we did not collect information about how many times participants had been exposed to information about food fraud, how long ago they had been exposed, or what specific food industries were implicated in the food fraud information they had encountered. Each of these may be an important determinant of persistent changes in consumer response to information. Consumer behavior following repeated exposure to information about food fraud incidents may be nonlinear, which could result in significant increases in consumer distrust of food

labeling systems after observing a particular number of food fraud incidents. Future research should focus on teasing out the relationship between number of exposures and relevant outcomes, such consumers' subjective expectations of the trustworthiness of product labeling.

The temporal proximity of exposure to food fraud information may also impact consumer response. There is relatively little relevant research in this area, though there are two studies that have examined the effect of time on information response in the context of food safety [26] and controversial technologies [27]. (Dillaway, Messer, Bernard, & Kaiser, 2011) find that the effect of a food safety event on consumers' WTP for chicken persists over time, while changes in WTP detected immediately after a single exposure to information about the use of controversial technologies, such as GMOs and meat irradiation, had eroded in a follow-up lab session [27]. There may also be an interaction effect between the number of exposures to food fraud information and the temporal distribution of those exposures on consumers' behavior. More work needs to be completed to understand differences in the effect of information on consumer decision-making over time.

Consumers may also respond differently to food fraud information based on the relevance of the product implicated in the scandal to them. In this research, we screened participants to ensure that they purchased and consumed olive oil, but we did not collect information about what products they had encountered food fraud information about. Participants who had previously read reports about the misrepresentation of seafood [28] may have valued olive oil differently in the pre-information round than participants who had read about olive oil mislabeling. More generally, it is possible that the probability that information about food fraud events affects one's long-term behavior may be affected by the relevance of the product category to the individual, perhaps because individuals are more likely to pay attention to information that is relevant to them [29].

A related issue that may have implications for these results is highlighted in a recent article. Based on survey results, Helen et al. [11] report that consumers develop strategies to avoid fraudulently labeled food products. In our experiment, we, the researchers, determined which olive oils the participants observed and valued. Imposing exogenously selected products on consumers permits identification of the effect of food fraud information on changes in consumer valuation but it does so only in the context of that exogenously selected product. If consumers were able to engage strategies they had developed that might involve substituting away from low-cost, imported bottles to more expensive but—subjectively—more trustworthy bottles, there would be a shift in consumption, but not necessarily a drop in quantity or value of olive oil purchased. We have two pieces of evidence that this may be a relevant consideration from our sample. First, there was not a significant difference in the average amount knowledgeable and unknowledgeable consumers spent per bottle on EVOO, suggesting that knowledgeable consumers may have adopted strategies to avoid mislabeled EVOO in the field. Secondly, participants believed that more domestically produced products are tested by the federal government than imported products, which would likely reduce their subjective expectation of the amount of mislabeling in domestic products. Additionally, [9] find that while food fraud information about olive oils from one country spills over onto consumer WTP for olive oils from other countries, domestically produced olive oil experiences the lowest drop in value. However, an experiment that provides room for consumers to respond strategically to negative information may yield additional important insights into the effects of food fraud.

This article is an initial step in determining important factors affecting consumers' response to food fraud scandals. Our findings indicate that prior exposure to information about food fraud is an important factor in explaining consumer behavior in the presence of food fraud. Future research in this direction can further refine estimates of the effect of exposure to food

fraud information and identify conditions that influence the effect of information on consumer behavior.

This study shows that prior knowledge of food fraud is related to baseline valuation of a product category—though not a specific brand—that has been implicated in food fraud scandals and to updating in response to new information. We have also identified several future directions of research on consumer behavior in the presence of food fraud that should be pursued. Although results show that consumers with prior knowledge of food fraud are less reactive to olive oil fraud information—because it appears that they may have already accounted for the possibility of food fraud occurring, we need further research to explore why they respond differently. Furthermore, we need further studies to understand how consumers' response to food fraud will change over time and with repeated exposure to information about food fraud incidents.

## Appendix

### Article on Italian olive oil fraud

**Italian olive oil industry and labeling scandals.**    Italy, along with Greece, Spain, and the U.S., is one of the leading global producers of olives and olive oil. Italy accounts for 16% of the total olive production in the world. The annual per capita olive oil consumption in Italy is approximately 12.35 KG. (FAOSTAT, 2013). The top Italian olive oil production regions are Puglia, Tuscany, Umbria, and Liguria.

While Italy is famous for olive oil, there have been allegations that Italian olive oil producers do not adhere to olive oil labeling standards, labeling olive oils as extra virgin that do not meet the standards that define extra virgin olive oils. Over 10% of Italian brands commonly sold in the U.S. market failed repeatedly to meet extra virgin olive oil standards ("Evaluation of Extra Virgin Olive Oil Sold in California", UC Davis, 2011). In 2015, Italian anti-fraud authorities investigated top Italian olive oil companies for mislabeling. They found that 9 out of the 20 largest brands mislabel low-quality olive oil as extra-virgin olive oil ("Italian olive oil scandal: top brands 'sold fake extra-virgin´´´, The Telegraph, November 11, 2015). Moreover, some producers use chemicals to cover up bad quality oils. It is not the first time that the Italian olive oil industry has come under scrutiny for fraud. In 2012, Italian anti-fraud authorities also found that the largest Italian olive oil producer had mislabeled domestic high-quality extra virgin olive oil with less expensive imported olive oil (The Guardian, November 11, 2015).

Italy is one of the top exporters of olive oil in the world. However, since the intensity and frequency of olive oil fraud has been on the rise, Italy faces a damaged reputation and economic losses. For this reason, Italy has established a special unit devoted to food fraud.

## Supporting information

**S1 File.**
(XLSX)

## Author Contributions

**Conceptualization:** Syed Imran Ali Meerza, Christopher R. Gustafson.

**Data curation:** Syed Imran Ali Meerza.

**Formal analysis:** Syed Imran Ali Meerza.

**Funding acquisition:** Christopher R. Gustafson.

**Methodology:** Syed Imran Ali Meerza.

**Project administration:** Syed Imran Ali Meerza.

**Resources:** Christopher R. Gustafson.

**Software:** Syed Imran Ali Meerza.

**Supervision:** Christopher R. Gustafson.

**Validation:** Syed Imran Ali Meerza, Christopher R. Gustafson.

**Writing – original draft:** Syed Imran Ali Meerza, Christopher R. Gustafson.

**Writing – review & editing:** Syed Imran Ali Meerza, Christopher R. Gustafson.

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
