## [Decision Letter · Decision Letter 0]

26 Sep 2019

PONE-D-19-23258

Prior Knowledge of Food Fraud Affect Consumer Behavior? Evidence from an Incentivized Economic Experiment

PLOS ONE

Dear Dr. Meerza,

Thank you for submitting your manuscript to PLOS ONE. After careful consideration, we feel that it has merit but does not fully meet PLOS ONE’s publication criteria as it currently stands. Therefore, we invite you to submit a revised version of the manuscript that addresses the points raised during the review process.

The authors should make some minor revisions on their manuscript according to the comments of both reviewers before considering it for publication. Overall, the work has been appreciated by both reviewers.

We would appreciate receiving your revised manuscript by Nov 10 2019 11:59PM. To enhance the reproducibility of your results, we recommend that if applicable you deposit your laboratory protocols in protocols.io, where a protocol can be assigned its own identifier (DOI) such that it can be cited independently in the future. For instructions see: http://journals.plos.org/plosone/s/submission-guidelines#loc-laboratory-protocols

We look forward to receiving your revised manuscript.

Kind regards,

Alberto Antonioni, PhD

Academic Editor

PLOS ONE

Journal Requirements:

3.  Please provide additional details regarding participant consent. In the ethics statement in the Methods and online submission information, please ensure that you have specified (1) whether consent was informed and (2) what type you obtained (for instance, written or verbal, and if verbal, how it was documented and witnessed). If the need for consent was waived by the IRB, please include this information.

4. Thank you for including your ethics statement:  "We received clearance from the University of Nebraska-Lincoln (UNL) Institutional Review Board to conduct this research with human subjects. The design of the experiment did not include deception of research subjects.

IRB# 20170616958 EX

Date of Approval: 04/10/2017".  

a.Please amend your current ethics statement to confirm that your named institutional review board or ethics committee specifically approved this study.

b.Once you have amended this/these statement(s) in the Methods section of the manuscript, please add the same text to the “Ethics Statement” field of the submission form (via “Edit Submission”).

5. Please ensure that you include a title page within your main document. We do appreciate that you have a title page document uploaded as a separate file, however, as per our author guidelines (http://journals.plos.org/plosone/s/submission-guidelines#loc-title-page) we do require this to be part of the manuscript file itself and not uploaded separately.

Additional Editor Comments (if provided):

Reviewers' comments:

Reviewer's Responses to Questions

**Comments to the Author**

1. Is the manuscript technically sound, and do the data support the conclusions?

Reviewer #1: Yes

Reviewer #2: Yes

2. Has the statistical analysis been performed appropriately and rigorously? 

Reviewer #1: Yes

Reviewer #2: Yes

3. Have the authors made all data underlying the findings in their manuscript fully available?

Reviewer #1: Yes

Reviewer #2: Yes

4. Is the manuscript presented in an intelligible fashion and written in standard English?

Reviewer #1: Yes

Reviewer #2: Yes

5. Review Comments to the Author

Reviewer #1: Prior Knowledge of Food Fraud Affect Consumer Behavior?

Evidence from an Incentivized Economic Experiment

PONE-D-19-23258

The authors use an experimental auction to examine how information about olive-oil fraud affects consumers’ willingness to pay (WTP) for both high-price and low-price olive oils, while also controlling for prior knowledge of food fraud incidents. Before being treated with information about food fraud, consumers with prior knowledge of fraud are willing to pay less for olive oil than those without prior knowledge of fraud. After being treated with information, both types of consumers reduce their WTP for olive oil, but those with prior knowledge reduce their WTP by less.

The study is well designed and the paper is well written. I have only minor comments.

1) On page 2 you say, “the total number of food fraud incidents in the two years from 2011 to 2012 was 60 percent higher than that in the three decades between 1980 and 2010.” That’s so stunning, I just want to be clear that I have it right. Does that mean the TOTAL number of incidents reported in that two-year period was 60 percent greater than the TOTAL number of incidents reported in the 30-year period preceding it? Or are you referring to the number of incidents reported per year?

2) I don’t understand the long sentence that begins the last paragraph on page 4. In particular, what do you mean when you say, “rather than simply resulting in repeated responses to new information about food fraud incidents”?

3) I don’t understand how to interpret the last row in Table 2. Does this mean knowledgeable subjects are only willing to $0.355 for olive oil? If so, which kind of oil – low or high priced? And is this figure referring to WTP before or after being treated with information? Unrelated to that, the third decimal place seems superfluous.

4) In Table 3 please indicate the number of people who came into the experiment with no prior knowledge and with prior knowledge, perhaps be including (N=X) and (N=Y) at the top of each column.

5) I don’t understand how you could’ve run the Hausman test you describe in the last sentence of Section 3. My understanding is that the test statistic is calculated by running the panel regression both ways (i.e., using fixed effects and random effects). But since Priorknow_fraud is individual invariant, it would just be absorbed by the individual-specific error terms in a fixed-effects regression. Please clarify.

6) Describing the results of regressions with interaction terms is notoriously tricky. By and large, you do a fine job. However, I think you’re wrong when you say, “After receiving information about Italian olive oil fraud, consumers’ valuations for Italian EVOO, on average, decreased by $4.42.” This is true only for uninformed consumers bidding on low-price oil. Uninformed consumers bidding on high-price oil saw a larger decrease. Namely, $4.42 + $4.65 = $9.07. Likewise, the change in WTP due to being treated with information were different for informed consumers.

7) Which type of oil do the bars in Figure 1 refer to? High-price oil, low-price oil, or an average of the two?

8) Are the $2.09 and $0.18 differences you report in connection with Figure 1 statistically meaningful? Based on the results from Table 4, the $2.09 difference clearly is. But I’m not sure about $0.18. If I were to guess, I’d say $0.18 isn’t statistically meaningfully different from zero. If I’m right, that’s a result that supports your argument. Before being treated with information, informed consumers bid significantly less than uninformed consumers. After being treated with information, there’s no meaningful difference in WTP.

9) That last paragraph from Section 3 probably belongs in Section 4.

10) You refer to the market for lemons on pages 16 and 17. Does this imply a greater role for third parties (government or otherwise)? Spence’s solution to Akerlof’s lemon problem is to introduce signals of quality.

11) I didn’t realize until I got to the top of page 19 that your “prior knowledge of fraud” variable referred to prior knowledge of ANY kind of food fraud. I’d been assuming you were referring specifically to prior knowledge of fraud in the olive-oil industry. Please be explicit about this earlier in the paper.

12) Throughout the paper, you’re using “informed” and “information” to refer to two different things. The first refers to information acquired before the experiment and the second refers to the information people are treated with during the experiment. This is confusing. Consider using a different term in place of “informed.” “Knowledgeable” would be one option.

Reviewer #2: This article examines the impact of prior knowledge about food fraud (in labeling) on consumer willingness to pay for Olive Oil. The authors find that consumers with prior knowledge of food fraud have a lower WTP as a baseline, and therefore respond less to an information treatment about food fraud. The paper is overall well written and the experimental methods are sound. The manuscript could just use some tightening and clarity.

The paper starts in a place that assumes a reader knows about, and cares about, food fraud. The introduction may be an opportunity to educate readers about what food fraud is – I recommend including some media-worthy recent examples. The evidence provided on the magnitude of the problem (i.e. number of large food frauds per year) could be moved up to the first paragraph (I was looking for that information early on). The severity of the problem for consumers could also be addressed – have cases of food fraud led to illness?

Did you ask specifically about food fraud in the EVOO market? If so, are there differences in ‘informed consumers’ vs. ‘informed about EVOO issues consumers’? From Table 2 it appears you asked familiarity as a (y/n) question, correct? Why not a Likert or other scale? Wording of the actual question familiarity question would be helpful.

Computer interface? (note on what you used)

On page 11, you could improve the link between Table 3 and the text. It feels repetitive to restate your control variables after the table, and yet because the variable names are new (and look like STATA code names), I felt that I needed to look again at the table.

Given the difference in age across the knowledge and no knowledge groups – please consider whether an interaction variable may be appropriate.

I think you missed the opportunity for some additional analysis that may be of interest:

-You indicated you had information about where they purchased olive oil. To the industry, I think it would be interesting to see if place of purchase was related to prior fraud knowledge and/or engagement with the information treatment. Do people seek out specialty stores? Do people who buy from specialty stores have a higher WTP because they assume those stores have done their homework on the products they sell, and therefore are less moved by any additional information?

- Is the WTP of prior knowledge group and the WTP of the no-knowledge group after information label statistically different or the same? You have reported the WTPs but I remain unclear whether these are statistically different. [Figure 1, error bars may answer my question?]

6. PLOS authors have the option to publish the peer review history of their article (what does this mean?). If published, this will include your full peer review and any attached files.

Reviewer #1: No

Reviewer #2: No

---

## [Author Response · Author response to Decision Letter 0]

8 Oct 2019

I have uploaded 'response to reviewers' and 'cover letter' to address all concerns raised by the reviewers and the Editor. Thank you.

---

## [Decision Letter · Decision Letter 1]

30 Oct 2019

Does Prior Knowledge of Food Fraud Affect Consumer Behavior? Evidence from an Incentivized Economic Experiment

PONE-D-19-23258R1

Dear Dr. Meerza,

We are pleased to inform you that your manuscript has been judged scientifically suitable for publication and will be formally accepted for publication once it complies with all outstanding technical requirements.

With kind regards,

Alberto Antonioni, PhD

Academic Editor

PLOS ONE

Additional Editor Comments (optional):

The authors satisfactorily addressed all comments by both reviewers. The current version of the manuscript can be now considered for publication.

Reviewers' comments:

Reviewer's Responses to Questions

**Comments to the Author**

1. If the authors have adequately addressed your comments raised in a previous round of review and you feel that this manuscript is now acceptable for publication, you may indicate that here to bypass the “Comments to the Author” section, enter your conflict of interest statement in the “Confidential to Editor” section, and submit your "Accept" recommendation.

Reviewer #2: All comments have been addressed

2. Is the manuscript technically sound, and do the data support the conclusions?

Reviewer #2: Yes

3. Has the statistical analysis been performed appropriately and rigorously? 

Reviewer #2: Yes

4. Have the authors made all data underlying the findings in their manuscript fully available?

Reviewer #2: Yes

5. Is the manuscript presented in an intelligible fashion and written in standard English?

Reviewer #2: Yes

6. Review Comments to the Author

Reviewer #2: I am satisfied with the revisions. The authors sufficiently address comments from both reviewers, and paid particular attention to comments mentioned by both reviewers.

7. PLOS authors have the option to publish the peer review history of their article (what does this mean?). If published, this will include your full peer review and any attached files.

Reviewer #2: No

---

## [Editor Report · Acceptance letter]

19 Nov 2019

PONE-D-19-23258R1 

Does Prior Knowledge of Food Fraud Affect Consumer Behavior? Evidence from an Incentivized Economic Experiment 

Dear Dr. Meerza:

I am pleased to inform you that your manuscript has been deemed suitable for publication in PLOS ONE. Congratulations! Your manuscript is now with our production department. 

With kind regards,

on behalf of

Dr. Alberto Antonioni 

Academic Editor

PLOS ONE